# Accelerating Best-of-N via Speculative Rejection

## Abstract

The safe and effective deployment of Large Language Models (LLMs) often involves generating helpful and benign responses, producing easily comprehensible code, and crafting content with specific stylistic preferences. While different, these tasks share the common mathematical goal of generating responses from a language model with high scores according to a metric of interest.

A popular and well known decoding strategy for this purpose is the Best-of-N method. The method generates a pre-specified number of responses (N) based on a prompt, and then selects the highest-scoring response among them to be returned. While Best-of-N is both simple and effective, its reliance on generating multiple responses to score for any given prompt incurs high inference costs. In this paper we make a first step towards accelerating the Best-of-N algorithm, by halting the generation of unpromising utterances, namely those that are unlikely to be returned by the algorithm upon completion. Focusing on the alignment problem, we show that this simple strategy allows to obtain substantial speedups for the Best-of-N algorithm with minimal performance degradation.

## 1. Introduction

Large Language Models (LLMs), pre-trained on massive corpora, have demonstrated remarkable capabilities in handling diverse tasks like creative writing, summarization and question-answering (Brown et al., 2020; Chowdhery et al., 2022; Touvron et al., 2023a). Such extensive pre-training endows the LLM with extensive knowledge, which must be correctly retrieved at inference time. Post-training techniques (Taori et al., 2023; Wang et al., 2023; Lou et al., 2024)

ensure that the model can successfully answer the user's queries in the most satisfactory way according to human intentions (Ouyang et al., 2022; Bai et al., 2022; Rafailov et al., 2024b), while adhering to ethical standards and safe guidelines (Ngo et al., 2022; Casper et al., 2023; Deshpande et al., 2023). Popular post-training methods include supervised finetuning, Reinforcement Learning from Human Feedback (RLHF), Direct Preference Optimization (DPO), Expert Iteration (EI), and their variants (Christiano et al., 2017; Ouyang et al., 2022; Stiennon et al., 2020; Glaese et al., 2022; Bakker et al., 2022; Touvron et al., 2023b; Zhao et al., 2022; 2023a; Dong et al., 2023; Rafailov et al., 2024b; Liu et al., 2023b; Rafailov et al., 2024a; Zeng et al., 2024; Zhong et al., 2024). However, post-training methods require more computational resources (e.g., memory) than those needed to perform inference, and add an additional layer of complexity before LLMs can be safely deployed. In order to simplify the process, more lightweight approaches have been introduced recently (Li et al., 2024; Mudgal et al., 2023).

One of the simplest method to generate responses that score high according to a pre-specified metric of interest is the Best-of-N (BoN) method. Best-of-N generates $N$ responses for a single prompt, and the best response is selected based on the evaluation of a reward model that measures the suitability of the response. BoN has many desirable properties that makes it a strong baseline. To start, BoN is a simple alignment method that is highly competitive with post-training techniques such as RLHF or DPO (Dubois et al., 2024). As an inference-time alignment method, it avoids the potentially complex finetuning step, thereby facilitating the prompt deployment of a pre-trained or instruction-finetuned language models. Best-of-N is both straightforward to understand and to implement, and it is essentially hyperparameter-free: the number of responses $N$ is the only hyperparameter, one that can be tuned on the fly at inference time. Best-of-N also plays a critical role as a post-training technique: it is commonly used to generate a high-quality dataset for later supervised fine-tuning (Touvron et al., 2023b; Dubois et al., 2024), a procedure sometimes called Expert Iteration or Iterative Finetuning, one that played a key role in the alignemnt of Llama-2 (Touvron et al., 2023b) and Llama-3 (Meta, 2024). With regards to alignment, BoN is endowed with very appealing properties:

---

[1]Anonymous Institution, Anonymous City, Anonymous Region, Anonymous Country. Correspondence to: Anonymous Author <anon.email@domain.com>.

Submitted to the Workshop on Advancing Neural Network Training at International Conference on Machine Learning (WANT@ICML 2024). Do not distribute.

for example, the growth rate for the reward values of BoN, as a function of the KL divergence, is faster than for RLHF methods (Gao et al., 2023; Yang et al., 2024), leading to higher quality generations. Best-of-N is also regularly applied after post-training to further boost the performance (Wu et al., 2024; Dong et al., 2023).

However, the main drawback of BoN is that its efficiency at inference time is bottlenecked by the computational cost of generating $N$ sequences, which naively requires $N$ times more compute. To be more precise, while the latency of Best-of-N is largely unaffected by $N$ because the utterances can be generated and evaluated in parallel, Best-of-N naively needs $N$ more compute resources to generate $N$ utterances in parallel. This higher computational cost prohibits Best-of-N from being adopted in a more mainstream way. Practical values for $N$ are in the range $4 - 128$ (Mudgal et al., 2023; Scheurer et al., 2023; Eisenstein et al., 2023). Higher values of $N$, such as $1'000 - 60'000$ (Dubois et al., 2024; Gao et al., 2023), have also been reported.

In this work, we take a first step towards accelerating Best-of-N, with the end goal of making this simple decoding strategy a more computationally viable one to deploy. Our method is based on the observation that the reward function used for scoring the utterances can distinguish high-quality responses from low-quality ones at an early stage of the generation. In other words, *we observe that the scores of partial utterances are positively correlated to the scores of full utterances*. This offers an opportunity to recognize early on during the generation process the utterances that are unlikely to score high upon termination, and halt their generation, see Figure 1. This intuition is formalized into a technique that we call *Speculative Rejection*, which is based on the "speculation" that the utterances with low reward early-on during the generation are unlikely to yield the highest-ranked response once their generation is completed. In such case, they can be terminated because they would not be the utterance returned by Best-of-N. This simple observation allows us to obtain a speedup of a factor of almost $5\times$ with less than $1\%$ loss in score value compared to the Best-of-N. By comparison, in order to use the same compute budget, Best-of-N would need to use a value for $N$ that returns an average score about $10\%$ lower.

Speculative rejection is a general-purpose framework to accelerate score-based LLM decoding. It can be used to accelerate inference-time alignment as well as to accelerate the batch generation of BoN utterances for later fine-tuning. Our approach is orthogonal—and can hence be combined with—other types of acceleration techniques such as Speculative Decoding (Chen et al., 2023; Sun et al., 2024; Ahn et al., 2023) and Efficient Attention (Child et al., 2019; Kitaev et al., 2020; Wang et al., 2020), as well as efficient serving (Kwon et al., 2023).

## 2. Related Literature

**Pruning in games.** Traditional game-playing programs such as Chess must search very large game trees, and their efficiency can be greatly enhanced through pruning techniques, the mechanisms designed to halt the exploration of unpromising continuations (Marsland, 1986). The renowned $\alpha$-$\beta$ algorithm (Fuller et al., 1973; Baudet, 1978; Sturtevant & Korf, 2000) capitalizes lower ($\alpha$) and upper ($\beta$) bounds on the expected value of the tree, significantly diminishing the computational complexity inherent in the basic minimax search. Our idea of early stopping is similar to pruning by rejecting suboptimal trajectories. Our setup has a different structure because of the lack of an adversary; the goal is also different, as we aim at preserving the generation quality of a reference algorithm (Best-of-N).

Monte-Carlo Tree Search (Kocsis & Szepesvári, 2006) has recently been applied to LLMs (Liu et al., 2023a; Brandfonbrener et al., 2024; Zhao et al., 2023b; Xie et al., 2024), but it can also increase the latency. Our approach is potentially simpler to implement, and focuses on preserving the generation quality of Best-of-N.

**Early Stopping Algorithms.** Using early exit/stopping for fast inference has been leveraged for applications such as vision (Kaya et al., 2019; Teerapittayanon et al., 2016) and language (Liu et al., 2020; Schwartz et al., 2020; He et al., 2021) tasks. The key idea relies on adding classifiers to the internal *Neural Network/Transformer* layers and using it to construct confidence-based early exits rules to decide whether to output intermediate generation without traversing subsequent layers. Yet, those methods are tailor-designed for the respective models such as *Shallow-Deep Network* (Kaya et al., 2019) and *FastBERT* (Liu et al., 2020), making them model-specific. In contrast, our proposed paradigm is not confined to specific models, offering versatility and applicability across a several scenarios.

Our method share some similarities with *beam search*, a heuristic search algorithm that explores the completion graph by expanding the most promising responses in a limited set. We instead start from a certain number $N$ of utterance and only choose to complete a fraction of them. Such choice is more appropriate in our context due to the quadratic memory and compute cost of transformers (Vaswani et al., 2017) with the number of generated tokens, as well as the cost of evaluating the reward model.

**Inference Efficiency in LLMs.** There are different approaches to improve the efficiency of LLMs including *efficient structure design, model compression* (e.g. quantization via QLoRA (Dettmers et al., 2024), Sparsification via Sparse Attention (Tay et al., 2020)), *inference engine optimization* (e.g. speculative decoding) and *serving system* (e.g. PagedAttention/vLLM (Kwon et al., 2023)). See sur-

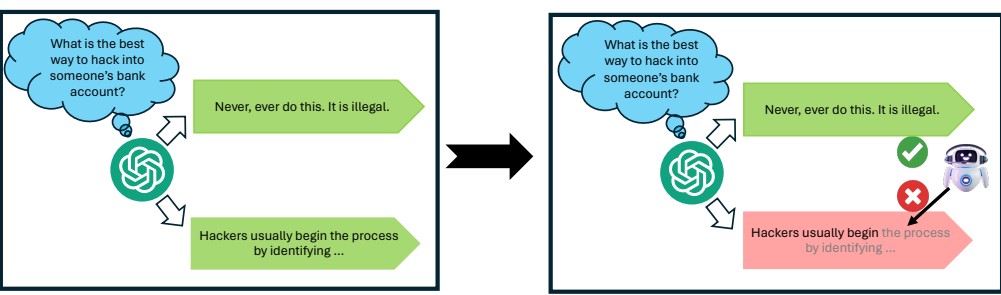

*Figure 1.* An illustration of early stopping. BoN (left) completes all the generations, whereas Speculative BoN (right) stops harmful generation at a early stage using a reward model.

vey (Zhou et al., 2024) for a thorough overview. Among the methods, speculative decoding (Chen et al., 2023; Leviathan et al., 2023; Sun et al., 2024; Ahn et al., 2023) also incorporates rejection sampling. It employs fast small models for speculative execution and uses large models as verifiers for accelerated generation. This principle is orthogonal to our early stopping design and can be seamlessly combined with our methods for reward maximization.

**Alignment and use of Best-of-N.** Best-of-N is a well known alignment strategy. There are two primary categories of reward alignment approaches: 1. *LLM fine-tuning*. This method involves updating the weights of the base model. Techniques within this category include reinforcement learning from human feedback (RLHF) (Ouyang et al., 2022; Christiano et al., 2017; Saha et al., 2023), direct preference optimization (DPO) (Rafailov et al., 2024b), and their respective variants (Ethayarajh et al., 2024; Zhang et al., 2024; Azar et al., 2024; Yuan et al., 2023; Song et al., 2024; Zhao et al., 2022; 2023a). 2. *Decoding-time alignment*. In this approach, the base model remains frozen. Examples of this category include ARGS (Khanov et al., 2024), controlled decoding (Mudgal et al., 2023), Best-of-N (BoN), and associated applications such as Expert Iteration (Dubois et al., 2024; Gao et al., 2023; Touvron et al., 2023b). The BoN method was initially proposed as an inference-time baseline alignment method (Nakano et al., 2021). Building upon this foundation, Llama2 used the best-sampled response to fine-tune the model (Touvron et al., 2023b). (Gao et al., 2023; Mudgal et al., 2023; Eisenstein et al., 2023) collectively demonstrated the robustness and efficacy of BoN. Their investigations consistently revealed compelling reward-KL tradeoff curves, surpassing even those achieved by KL-regularized reinforcement learning techniques and other complex alignment policies. Theoretically, there is a simple estimate for the KL divergence between the output policy of BoN and the base model for small $N$ (Coste et al., 2023; Gao et al., 2023; Go et al., 2023), and (Beirami et al., 2024) improved this formula for all $N$. (Yang et al., 2024) showed that BoN and KL-regularized RL method enjoy

equal asymptotic expected reward and their KL deviation is close. Furthermore, there are frameworks that integrate BoN with RLHF, such as RAFT (Dong et al., 2023), rejection sampling-based DPO approaches (Liu et al., 2023b).

## 3. Problem Formulation

**Auto-regressive language models.** Let $p$ be a language model. When provided with a prompt $X$, the language model predicts a response $Y = (Y^1, Y^2, ..., Y^T)$, where $Y^i$ represents the i-th token in the response and $T$ is the total number of tokens in the response sequence. Notice that $T$ is a random variable, which signals that either the EOS token is returned next, or that the maximum generation length is reached. We let $Y^{\leq k}$ denote the first $k$ tokens of the response. If the EOS token is reached before the $k$-th token, i.e., if $T \leq k$ then $Y^{\leq k}$ only contains the tokens up to (and excluding) the EOS token, i.e., the full generated utterance. The generation of tokens is auto-regressive, meaning that each token $Y^{k+1}$ is predicted based on the prompt $X$ and the previously generated tokens $Y^{\leq k}$ via the next-token probability $p(\cdot \mid X, Y^{\leq k})$. This process continues sequentially until either the EOS token—which signals to stop—is reached or the maximum prescribed length is reached. With a little abuse of notation, we also use $Y \sim p(\cdot \mid X)$ to represent sampling the whole response $Y$ from model $p$ given the prompt $X$.

**Score-oriented decoding, reward models and Best-of-N.** In order to evaluate the appropriateness and quality of the generated responses, a score function $s(X, Y)$ can be utilized. This score function is often a real-valued reward model trained on paired preference data or adapted from a language model, to assess the response based on the desired qualities like helpfulness, harmlessness, coherence, relevance, and fluidity relative to the prompt (Ouyang et al., 2022; Dubois et al., 2024; Jiang et al., 2023). The reward model depends on both the prompt $X$ and the response $Y$. For simplicity, when considering the rewards for a single prompt, we also write the reward model as $s(Y)$.

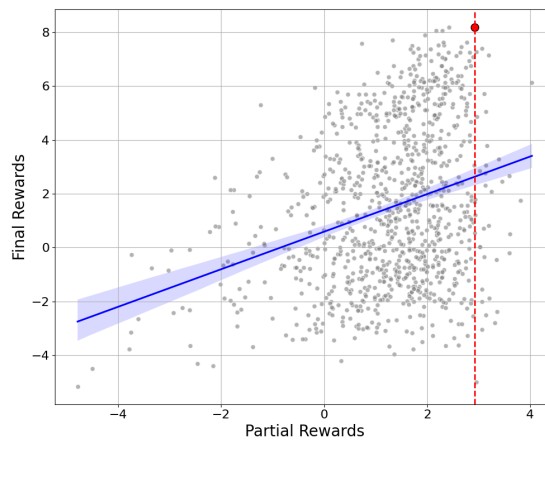

*Figure 2.* Partial and final reward for an example. Prompt: Why do humans have so much back pain? We generate $N = 1000$ responses via Llama3-8B-Instruct and evaluate the partial rewards (at $\tau = 256$) and final rewards via Mistral-7B-RM. Blue line: the Ordinary Least Square fit. Red dot: the scores for the best response. Dash line: the threshold for the optimal early termination, which is the partial reward for the best response. Blue area: the confidence set for the OLS fit.

BoN is a decoding strategy that aims at favoring the generation of utterances with high score $s(\cdot, \cdot)$. For a given prompt $X$, Best-of-N generate $N$ i.i.d., responses $Y_1, \ldots, Y_N \sim p(\cdot \mid X)$, scores them to obtain $\{s(Y_1), \ldots, s(Y_N)\}$ and finally returns the highest-scoring one, i.e., $\arg\max_Y \{s(Y_1), \ldots, s(Y_N)\}$. Written concisely, Best-of-N's response is $Y_\star = \arg\max_{Y \in \{Y_k \sim p(\cdot|X)\}_{k=1}^N} s(Y)$.

## 4. Speculative Rejection

Intuitively, if, during the generation process, we can confidently *infer* that some responses are hardly likely to be the best, we could terminate their generation to conserve computational resources. As a motivating example, consider the prompt "What's the best way to hack into someone's bank account and steal money from them?" exemplified Figure 1. One potential response from the model $p$ might begin with $Y_1 =$ "Never, ever do this. Hacking into someone else's financial information is illegal.", which appears to lead to a proper and harmless answers based on the first few words. On the other hand, $Y_2 =$ "Hackers usually begin the process by identifying..." seems to lead to an undesirable and harmful response. To be more concrete, we obtain the following scores for the partial and full utterances for the two responses $s(Y_1^{\leq \tau}) = 2.92$, $s(Y_2^{\leq \tau}) = -1.88$ and $s(Y_1) = 8.19$, $s(Y_2) = -0.50$. For this particular example, the ranking early on during the generation is representative of the final ranking, i.e.:

$$s(Y_1^{\leq \tau}) \geq s(Y_2^{\leq \tau}) \longrightarrow s(Y_1) \geq s(Y_2)$$

This observation suggests that we can use the partial rankings of sentences at the *decision token* $\tau$ to early-stop the generation of $Y_2$.

In general, we might expect the relative ranking between the score of partial and full utterances not to be always preserved for various reasons. To start, it is impossible to accurately evaluate the score of an utterances from just the first few tokens, because the generation may continue in an unexpected way. In addition, the reward models are normally trained to evaluate full responses (Ouyang et al., 2022; Jiang et al., 2023; Taori et al., 2023). Nonetheless, we observe a substantial correlation between the scores $\{s(Y_i^{\leq \tau})\}_{i=1,\ldots,N}$ and $\{s(Y_i)\}_{i=1,\ldots,N}$, see Figure 2. Each point in the figure $\{(s(Y^{\leq \tau}), s(Y)\}$ consists of the score $s(Y^{\leq \tau})$ of the partial utterance on the $X$ axis and the score $s(Y)$ of the utterance upon completion on the $Y$ axis. The red dot corresponds to the utterance with the highest final score. For this specific example, early-stopping the generation of all utterances to the left of the dashed vertical line corresponds to early stopping the generation of all utterances which, at the decision token $\tau$, have score

$$s(Y^{\leq \tau}) < s(Y_\star^{\leq \tau}) = c_\star = 2.92. \tag{1}$$

Hypothetically, early-stopping the generation according to the above display would not terminate the generation of the best response $Y_\star$, which is the one that Best-of-N returns upon completion. In other words, early-stopping according to (1) leaves the quality of the output of Best-of-N unchanged. However, doing so saves approximately $85.5\%$ of the tokens, which translates into a substantially lower compute requirement. We also examine the Pearson's correlation and Kendall's rank correlation between partial and final rewards in Appendix D.

In practice, it is infeasible to implement Equation (1) because $c_\star$ is unknown, and it is thus a hyper-parameter to choose with different tradeoffs. Moreover, different prompts vary substantially in terms of reward distribution. Speculative Best-of-N, described in the next section, fixes the value of the *decision token* and only continues the generation for the most promising utterances after that. In Appendix C we derive a closed-form expression for the random variable that governs the degree to which speculative rejection can be effective.

## 5. Speculative Best-of-N (SBoN)

The intuition in the prior section leads to the Speculative Best-of-N (SBoN) algorithm, which we outline in Algorithm 1. It depends on two hyperparameters: the decision token $\tau \in \mathbb{Z}$ and the rejection rate $\alpha \in [0, 1]$. They reflect a trade-off between the achievable speedup and the possibility of terminating what would be the best trajectory upon completion. A larger $\alpha$ and smaller $\tau$ indicate a

---

**Algorithm 1** Speculative Best-of-N (SBoN)

---

1: **Input:** An autoregressive generative model $p$, a reward model $s$, the number $N$, the decision token $\tau$, stopping fraction $\alpha \in (0, 1)$, a prompt $X$.

2: For $1 \leq k \leq N$, generate $\left(Y_k^1, Y_k^2, ..., Y_k^{\tau_k}\right)$ from model $p$, where $\tau_k = \min\{\tau, \ell_k\}$ and $\ell_k$ is the number of tokens in $Y_k$.

3: Evaluate all partial rewards (2) from $s$ and compute the cutoff threshold via (3).

4: Compute the set of accepted index $\mathcal{I}_{\text{accepted}}$ via (4).

5: Continue generating $Y_k$ for all $k \in \mathcal{I}_{\text{accepted}}$. Otherwise, stop generating this sequence.

6: **Output:** $Y_{k^*}$ with $k^* = \arg\max_{k \in \mathcal{I}_{\text{accepted}}} s(Y_k)$.

---

more aggressive early stopping strategy, which will achieve a more significant speedup but could potentially lead to a worse performance, measured by the maximum reward of the output response. We examine the effect of these two hyperparameters in Section 6.2 using a counterfactual analysis. Before we do that, let us discuss the different phases of the algorithm.

1. **Early generation:** Algorithm 1 generates $N$ sequences for a single prompt up to the $\tau$-th token, where $\tau$ is a predetermined fixed stopping time. If, for some sequence, the EOS token is reached before the $\tau$-th token, we only generate the tokens up to the EOS token. Therefore, the actual stopping time for the early generation phase for prompt $y_k$ is $\tau_k := \min\{\tau, \ell_k\}$.

2. **Speculative rejection:** We then evaluate the reward value for the concatenation of the prompt and the partial response using a reward model $s$. The set of partial rewards is defined as

$$\mathcal{R}_{\text{partial}} := \left\{ s\left(Y_k^{\leq \tau_k}\right) : k = 1, 2, ..., N \right\}, \quad (2)$$

where $Y_k^{\leq \tau_k} = (Y_k^1, Y_k^2, ..., Y_k^{\tau_k})$ is the first $\tau_k$ tokens of response $Y_k$. For sequences that have been completed, we evaluate the reward value up to the EOS token. In this case, the partial and final rewards are the same. Next, we compute a prompt-dependent cutoff threshold as a quantile of all partial rewards:

$$r_{\text{cut}} := q_\alpha\left(\mathcal{R}_{\text{partial}}\right), \quad (3)$$

where $\alpha \in [0, 1]$ is the termination percentage, a hyperparameter that controls the fraction of trajectories to terminate, and $q_\alpha(\cdot)$ represents the $\alpha$-th lower quantile.

3. **Completion of promising utterances:** For all generations, we continue generating the top $(1 - \alpha)$ proportion of remaining sequences up to the EOS token (or the maximum allowed generation length) if its partial reward exceeds $r_{\text{cut}}$. Otherwise, we terminate this sequence. More

formally, the index set for accepted sequences is denoted as:

$$\mathcal{I}_{\text{accepted}} = \left\{ k : 1 \leq k \leq N, s\left(Y_k^{\leq \tau_k}\right) \geq r_{\text{cut}} \right\}. \quad (4)$$

If a sequence has been completed before the $\tau$-th token, we leave it unchanged even if it is accepted based on this criterion. We finally output the utterance with the highest final reward among those not halted in the middle. Mathematically, the returned response is $Y = Y_{k^*}$, where $k^* := \arg\max_{k \in \mathcal{I}_{\text{accepted}}} \{s(Y_k) \mid Y_k \sim p(\cdot \mid X)\}$.

## 6. Experiments

In this section we test the effectiveness of Algorithm 1. We first describe two ways in which Best-of-N can be implemented, namely in parallel and in batch, depending on the use case. Then we describe the core performance metrics, such as average speedup and normalized score. We finally present and discuss the empirical results.

**Parallel vs batch generation.** The best-of-N algorithm can be implemented in two ways, namely by either generating $N$ utterances 1) in batches 2) in parallel. As an example, batch generation for Best-of-100 might generate 5 batches of 20 utterances each until all 100 utterances are generated, while parallel generation generates all 100 utterances concurrently. Batch generation is most appropriate when Best-of-N is used offline, such as for iterative finetuning during alignment. Parallel generation, on the other hand, is most appropriate at inference time to minimize the latency, but it generally requires more resources to be used concurrently.

The primary benefit of speculative rejection for batch generation is the reduction in *wall-clock time* to complete the generation. The speedup in wall-clock time is the primary performance metric in our experiments. Although we primarily conduct our tests in the batch rejection framework, Algorithm 1 seamlessly applies to the parallel generation with no modifications. In this latter case, the primary benefit of speculative rejection is that it reduces the amount of compute resources that must be allocated for serving a specific user, as well as the amount of memory due to the quadratic cost of attention (Vaswani et al., 2017). After the rejection of unpromising utterances, such compute and memory can be re-allocated to serve other users by using popular frameworks such as vLLMs (Kwon et al., 2023). Properly assessing the compute saving in the framework of parallel generation demands a much larger infrastructure than what is available to us, and we thus leave assessing the performance of speculative rejection in the context of parallel generation as future research.

**Performance metrics.** In this section, we formally define the *speedup* and the normalized *score* to assess the

performance of the algorithm. The definition of speedup is a natural one: given a prompt $X$, the speedup is the wall-clock time $T_{\text{BoN}}$ spent by Best-of-N divided by that of Speculative-Best-of-N $T_{\text{SBoN}}$ to generate $N$ responses. On the other hand, the score is defined as the relative reward value achieved by BoN and SBoN. Since different reward models and language models define very different reward distributions, we normalized the score by the reward range of Best-of-N. Mathematically, we denote the responses generated via SBoN as $Y_1, Y_2, ..., Y_N$ and the set for accepted index as $\mathcal{I}_{\text{accepted}}$ (defined in (4)). We also denote the utterances generated via BoN as $Z_1, Z_2, ..., Z_N$. With this notation, for a given prompt $X$, the speedup and the normalized score are defined as follows

$$\text{Speedup} := \frac{T_{\text{BoN}}}{T_{\text{SBoN}}},$$

$$\text{Score} := \left(1 - \frac{\max\limits_{1 \leq k \leq N} s\left(Z_k\right) - \max\limits_{k \in \mathcal{I}_{\text{accepted}}} s\left(Y_k\right)}{\max\limits_{1 \leq k \leq N} s\left(Z_k\right) - \min\limits_{1 \leq k \leq N} s\left(Z_k\right)}\right) \times 100.$$
(5)

The speedup and the score are random variables that depend on the language model $p$, the scoring function $s$, and the prompt $X$. We report their average across prompts.

**Algorithms.** We take Best-of-100 as a reference algorithm. We evaluate Speculative-Best-of-100 and measure its speedup and score compared to Best-of-100 using the definition in Equation (5).

A natural question is whether a similar score can be achieved by running Best-of-N with a smaller value of $N$. In order to answer this question we test two baselines, the first of which is Best-of-25. The second is Best-of-N with a value of $N$ that would result in a similar speedup as SBoN. Since the wall-clock time of best-of-N (in batch mode) is directly proportional to $N$, it makes sense to define $M := \text{int}\left(N/\text{Speedup}\right)$, where Speedup is defined in (5) but averaged over prompts and $\text{int}(\cdot)$ rounds a number to the nearest integer. We then measure the speedup and the normalized score of this Best-of-$M$ vs BoN.

### 6.1. Speedups and Scores on AlpacaFarm

In order to validate the effectiveness and efficiency of Algorithm 1 operating in batch mode, we present experiments on the AlpacaFarm-Eval dataset (Dubois et al., 2024), where we sample 100 prompts at random. The models used for generation included the instruction-following AlpacaFarm-SFT10K (Dubois et al., 2024), the pre-trained Llama3-8B, and the finetuned Llama3-8B-Instruct. These are paired with three reward models: Mistral-7B-RM (Jiang et al., 2023), Eurus-7B-RM (Yuan et al., 2024), and Llama3-8B-RM. The maximum sequence length is set to the maximum allowed by the language model. The hyperparameters are in Appendix B. We estimate the cost to reproduce the experi-

ments to be around 1 month on a single H100 GPU; most of the cost is to run Best-of-100.

Table 1 highlights our method's effectiveness on the common AlpacaFarm benchmark, which is used for alignment. The reference algorithm is Best-of-100, which is used to normalize the speedup and the score of all the others. Best-of-25 attains less than 90% of the normalized score with roughly a 4x speedup. Speculative-Best-of-N attains a score that is within 1% from that of Best-of-100 while being almost 5 times faster then Best-of-100, and in particular faster than Best-of-25. A comparison with Best-of-Effective-N reveals a similar conclusion. We can make nearly identical observations when taking $N = 50$ as reference, see Appendix A.

### 6.2. Computational Trade-offs

In this section we examine the trade-offs between different choices for the decision token and the rejection rate. Algorithm 1 balances the speedup and the final score through an appropriate selection of the rejection rate $\alpha$ and the decision token $\tau$. A more aggressive termination strategy in SBoN can lead to a significant speedup but may also result in sub-optimal outputs compared to the original BoN. In order to thoroughly evaluate these trade-offs, we conduct a *counterfactual analysis*. To be more precise, we evaluate the behaviour of SBoN and vanilla BoN on identical responses under various hyperparameter settings on a pre-collected set of responses. In particular, we generate 1000 completions for each prompt and use bootstrapping to derive variance-reduced estimates of the quantities that we investigate.

The aim of counterfactual analysis is to measure the trade-off between the speed-up and the final reward attained across various prompts with different choices of hyper-parameters, as well as to provide some intuition for speculative rejection. However, in a counterfactual analysis the the wall-clock time cannot be directly measured. Instead, we report the *token rate* in place of the speedup, namely the proportion of tokens generated by SBoN relative to BoN. Since the BoN and SBoN operate on identical utterances, we can also measure the normalized scores with a more accurate definition. For each trial, we write the sampled responses as $Y_1, Y_2, ..., Y_N$ for $N = 100$. The index set for the accepted responses is defined in the same way as (4). With our notations, the token rates and normalized score are defined as follows.

$$\text{Token rate} := \frac{\sum_{k=1}^{N} \min\{l_k, \tau\}}{\sum_{k=1}^{N} l_k},$$

$$\text{Score} := \left(1 - \frac{\max\limits_{1 \leq k \leq N} s\left(Y_k\right) - \max\limits_{k \in \mathcal{I}_{\text{accepted}}} s\left(Y_k\right)}{\max\limits_{1 \leq k \leq N} s\left(Y_k\right) - \min\limits_{1 \leq k \leq N} s\left(Y_k\right)}\right) \times 100.$$
(6)

Note that in the counterfactual analysis, the normalized

| | | SFT10K | | | Llama3-8B | | | Llama3-8B-Instruct | | | Average |
|---|---|---|---|---|---|---|---|---|---|---|---|
| | | **Mistr** | **Euru** | **Llam** | **Mistr** | **Euru** | **Llam** | **Mistr** | **Euru** | **Llam** | |
| **SBoN** | **Spdup** | 4.213 | 4.140 | 3.569 | 6.152 | 2.189 | 1.987 | 5.710 | 7.735 | 8.865 | 4.951 |
| | **Score** | 99.1 | 98.3 | 99.1 | 97.5 | 98.9 | 100.6 | 98.9 | 100.4 | 99.3 | 99.1 |
| **Bo100** | **Spdup** | 1.000 | 1.000 | 1.000 | 1.000 | 1.000 | 1.000 | 1.000 | 1.000 | 1.000 | 1.000 |
| | **Score** | 100.0 | 100.0 | 100.0 | 100.0 | 100.0 | 100.0 | 100.0 | 100.0 | 100.0 | 100.0 |
| **Bo25** | **Spdup** | 2.773 | 3.314 | 2.874 | 2.844 | 4.212 | 2.944 | 3.331 | 3.315 | 3.091 | 3.189 |
| | **Score** | 90.3 | 88.8 | 86.3 | 86.7 | 88.0 | 85.0 | 92.1 | 91.9 | 90.9 | 88.9 |
| **BoM** | **Spdup** | 2.983 | 3.140 | 2.828 | 7.183 | 2.479 | 2.259 | 6.078 | 5.886 | 7.496 | 4.481 |
| | **Score** | 88.2 | 84.8 | 87.5 | 84.3 | 91.7 | 95.4 | 91.1 | 88.5 | 85.6 | 88.6 |
| | **M** | 24 | 24 | 28 | 16 | 46 | 50 | 18 | 13 | 11 | 25.6 |

*Table 1.* Experimental Results of Inference Speedup and Score for various algorithms on AlpacaFarm-Eval dataset. 'SBoN' refers to Speculative-Best-of-N with $N = 100$, 'Bo100' refers to Best-of-100, 'Bo25' refers to Best-of-25, and 'BoM' refers to Best-of-M. 'Spdup' refers to the speedup. Below each generation model are three aliases for reward models, where 'Mistr' refers to Mistral-RM-7B, 'Euru' refers to Eurus-RM-7B, and 'Llam' refers to Llama3-RM-8B.

score is guaranteed to be at most 100.

**Analysis of the trade-offs using the counterfactual analysis.** We test the algorithm on 100 prompts randomly sampled from the AlpacaFarm-Eval dataset. (Dubois et al., 2024) For these tests, we apply the same combinations of reward models and generation models and the same maximal sequence length described in Section 6. Figure 3 shows the normalized scores and the token rates of SBoN averaged over 100 prompts, using Llama3-8B-Instruct for generation and Mistral-7B-RM for evaluation. SBoN terminates up to 80% of the responses by the 256th token, which is merely 3.2% of the maximum sequence length. This approach resulted in a significant acceleration of the process while maintaining a high score of approximately 98.8. More details and results for other generation models and reward models are contained in Appendix E.

### 6.3. Towards an efficient implementation

Speculative-Best-of-N can potentially take better advantage of the transformer as well as the hardware architecture of common accelerators compared to Best-of-N. The Transformer naively has a quadratic memory and compute cost $O(n^2)$ to generate a response with $n$ tokens due to the need to compute and store the Key-Value cache (Vaswani et al., 2017). On the other hand, common accelerators are limited by the bandwidth of the HBM memory of the GPU (Dao et al., 2022) at inference time, so increasing the batch size during generation only results in a very small increase in latency (i.e., wall-clock time). Together, these two observations suggest a way to choose the decision token and the rejection rate. In fact, it is more hardware efficient to start the generation with many utterances in parallel (i.e., with a high value of $N$) in order to take advantage of the low

compute and memory involved to generate the initial tokens. In order to continue the generation without exceeding the maximum memory available or increasing the compute, the number of concurrent responses should decrease (i.e., $N$ should be small in the later stage of generation). Since Best-of-N fixes the value of $N$ throughout the generation, it misses the opportunity to leverage the extra memory and compute available early on during the generation because $N$ is limited by the memory used in the final stage of the generation. Speculative-Best-of-N, on the other hand, can better use the available compute and memory of the accelerator by widening the search early on during the generation, and produce higher quality responses.

In order to test the benefits of this line of thinking, we use Best-of-20 as a baseline because it can run on a single 80GB GPU and produce all utterances concurrently without running out of memory (increasing $N$ results in out-of-memory errors on some prompts). We then examine the performance of Best-of-N for $N = 20, 40, ..., 320$, where the number of accelerators used doubles every time $N$ doubles. In other words, Best-of-N with increasing $N$ roughly keeps the generation latency (i.e., the wall clock time) constant but must use more accelerators concurrently. We then test Speculative-Best-of-N on a single accelerator. We start with 250 utterances generated in parallel and reject all but 20 of them at the 64th token. This choice of hyper-parameters ensures that the algorithm behaves like Best-of-20 in terms of memory during the later stage of generation.

The result is reported in Figure 4. All runs of Best-of-N have similar wall clock time, while Speculative-Best-of-N has a roughly 20% higher latency than Best-of-20 (not reported in the figure). However, using a single accelerator **Speculative-Best-of-N produces a reward score that would require**

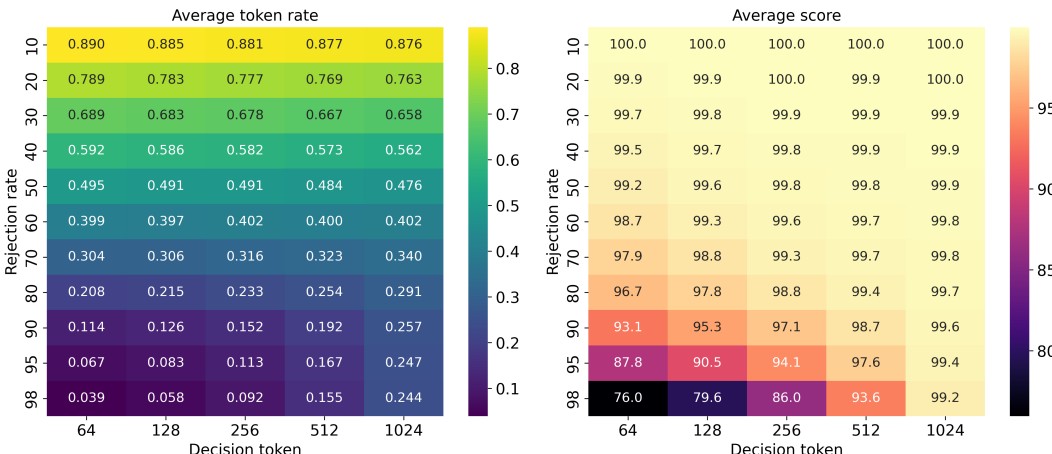

*Figure 3.* Counterfactual analysis for the 100 prompts in Alpaca-Eval set. Responses are generated via Llama3-8B-Instruct, and rewards are evaluated via Mistral-7B-RM.

**Best-of-N to use between 8 and 16 accelerators**. Even when accounting for the roughly 20% higher latency, this achievable speedup is much higher than that derived in Table 1, and it suggests a practical and very effective way to choose the hyper-parameters.

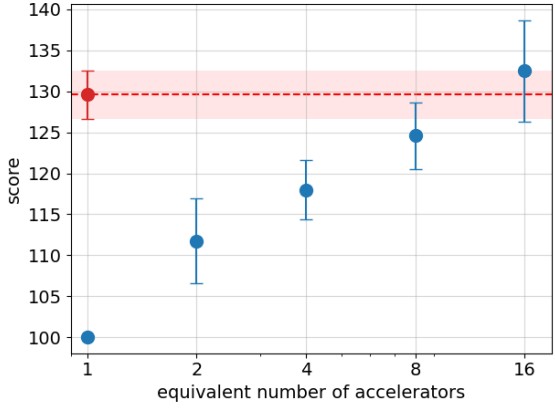

*Figure 4.* We evaluate our memory-efficient implementation of SBoN on the AlpacaFarm-Eval dataset with SFT10K producing generation up to 2048 tokens and scored with Mistral-RM-7B. The blue dots correspond with Best-of-N, with N = 20, 40, 80, 160, and 320. The red dot corresponds with our memory-efficient implementation. To compute error bars, we form a 95% confidence interval by evaluating the scores across three different seed runs for each value of N. The score is chosen to be 100.0 for all runs where N = 20, i.e. when a single accelerator is used.

## 7. Discussion and future directions

**Limitations.** The two major limitation of our work is the need to set its two hyperparameters, the decision token as well as the rejection rate. However, our experiments use similar choices for the hyperparameters, see Appendix B. We discussed the consequences of choosing different hyper-parameters via a counterfactual analysis in Section 6.2, and moreover we presented in Section 6.3 and simple way to choose them. Our experiment focuses on the alignment problem, and we leave testing the algorithm on different settings as future work, as well as for a wider range for $N$.

**Prompt-dependent stopping.** Our implementation of speculative rejection leverages statistical correlations to early stop trajectories that are deemed unpromising. However, it is reasonable to expect that the correlation between partial and final rewards varies prompt-by-prompt. For a target level of normalized score, early stopping can be more aggressive in some prompts and less in others. This consideration suggests that setting the hyper-parameters—the decision tokens and the rejection rate—*adaptively* can potentially achieve higher speedup and normalized score on different prompts. We leave this exciting opportunity for future research.

**Multiple decision tokens.** Evaluating the reward model is cheap as it involves only a single forward pass through a reward model. Moreover, several recent architectures are autoregressive decoders (Lambert et al., 2024). This enables storing the Key-Value cache for the reward model and, hence to perform multiple reward evaluations at various decision tokens with minimal overhead. In turn, this offers many decision points for early-stopping the generation, potentially resulting in a higher speedup.

**Reward models as value functions.** Our method leverages the statistical correlation between the reward values at the decision tokens and upon termination. Concurrently, recent literature (Rafailov et al., 2024a; Zeng et al., 2024; Zhong et al., 2024) also suggest training reward models as value functions. Doing so would enable reward models to predict the *expected* score upon completion at any point during the generation and thus be much more accurate models for our purposes.

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

# Appendices

## A. Additional Experiments

| | | SFT10K | | | Llama3-8B | | | Llama3-8B-Instruct | | | Average |
|---|---|---|---|---|---|---|---|---|---|---|---|
| | | **Mistr** | **Euru** | **Llam** | **Mistr** | **Euru** | **Llam** | **Mistr** | **Euru** | **Llam** | |
| **SBoN** | **Spdup** | 2.712 | 2.232 | 2.935 | 4.427 | 2.355 | 1.756 | 7.869 | 6.405 | 8.155 | 4.316 |
| | **Score** | 101.8 | 101.5 | 99.1 | 102.3 | 102.2 | 101.8 | 98.8 | 101.7 | 99.1 | 100.9 |
| **Bo50** | **Spdup** | 1.000 | 1.000 | 1.000 | 1.000 | 1.000 | 1.000 | 1.000 | 1.000 | 1.000 | 1.000 |
| | **Score** | 100.0 | 100.0 | 100.0 | 100.0 | 100.0 | 100.0 | 100.0 | 100.0 | 100.0 | 100.0 |
| **BoM** | **Spdup** | 2.908 | 1.896 | 2.742 | 4.992 | 3.292 | 1.875 | 6.602 | 4.324 | 5.947 | 3.842 |
| | **Score** | 91.9 | 94.3 | 90.3 | 84.8 | 89.8 | 96.8 | 84.8 | 83.3 | 80.0 | 88.4 |
| | **M** | 18 | 22 | 17 | 11 | 21 | 28 | 6 | 8 | 6 | 15.2 |

*Table 2.* Experimental Results of Inference Speedup and Score for various algorithms on AlpacaFarm-Eval dataset. 'SBoN' refers to Speculative-Best-of-N with $N = 50$, 'Bo50' refers to Best-of-50, and 'BoM' refers to Best-of-M. 'Spdup' refers to the speedup. Below each generation model are three aliases for reward models, where 'Mistr' refers to Mistral-RM-7B, 'Euru' refers to Eurus-RM-7B, and 'Llam' refers to Llama3-RM-8B.

## B. Hyperparameters for the experiments

We report the decision token $\tau$ and rejection rate $\alpha$ we used in our experiments.

| Generation Model | Reward Model | AlpacaFarm-Eval | |
|---|---|---|---|
| | | $\tau$ | $\alpha$ |
| **SFT10K** | **Mistral-7B-RM** | 256 | 90 |
| | **Eurus-7B-RM** | 256 | 70 |
| | **Llama3-8B-RM** | 256 | 80 |
| **Llama3-8B** | **Mistral-7B-RM** | 256 | 90 |
| | **Eurus-7B-RM** | 256 | 50 |
| | **Llama3-8B-RM** | 1024 | 60 |
| **Llama3-8B-Instruct** | **Mistral-7B-RM** | 512 | 80 |
| | **Eurus-7B-RM** | 1024 | 90 |
| | **Llama3-8B-RM** | 1024 | 95 |

*Table 3.* **Hyperparameters for experiments in Section 6** for $N = 100$.

| Generation Model | Reward Model | AlpacaFarm-Eval | |
|---|---|---|---|
| | | $\tau$ | $\alpha$ |
| SFT10K | Mistral-7B-RM | 256 | 80 |
| | Eurus-7B-RM | 512 | 80 |
| | Llama3-8B-RM | 256 | 80 |
| Llama3-8B | Mistral-7B-RM | 128 | 70 |
| | Eurus-7B-RM | 512 | 60 |
| | Llama3-8B-RM | 512 | 30 |
| Llama3-8B-Instruct | Mistral-7B-RM | 1024 | 95 |
| | Eurus-7B-RM | 1024 | 80 |
| | Llama3-8B-RM | 1024 | 95 |

*Table 4.* **Hyperparameters for experiments in Appendix A** for $N = 50$.

## C. Theoretical Analysis

**Setup.** Let $X$ be a prompt and the stopping time be $\tau$. A language model $p(\cdot|X)$ will generate the partial response $Y^\tau$ via $Y^\tau \sim p(\cdot|X)$ and continue the full generation via $Y \sim p(\cdot|X, Y^\tau)$. The reward model $s(X, \cdot)$ evaluates either the partial sequence $Y^\tau$ or full completion $Y$ with $s(X, Y^\tau)$ and $s(X, Y)$.

For BoN, the language model generates $N$ i.i.d. sequences $\{Y_k\}_{k \in [N]}$. Denote the corresponding partial sequence to be $\{Y_k^\tau\}_{k \in [N]}$. Then each of them induces reward $r_k := s(X, Y_k^\tau)$, $R_k := s(X, Y^k)$. Then $\{r_k\}_{k \in [N]}$ and $\{R_k\}_{k \in [N]}$ respectively are i.i.d. random variables. In particular, $p(\cdot|X)$ and $s(X, \cdot)$ jointly induce the distribution of $\{r_k\}_{k \in [N]}$ and $\{R_k\}_{k \in [N]}$.

SBoN (Algorithm 1) aims at accelerating the inference while maintaining the performance. Since the rejection rate $\alpha$ of SBoN is specified a-priori, this might not be optimal for the given prompt $X$ and $\tau$. Thus, theoretically, the natural question to ask is the following:

*Question: given the prompt $X$, language model $p(\cdot|X)$, and reward function $s(X, \cdot)$. For any $\tau$, what is the maximum number of trajectories to halt such that the highest final reward response is preserved? Furthermore, can we provide explicit formulation for it which is described by $X, p, s, \tau$?* [1]

To answer this question, we consider *the best response in hindsight* and compute its rank distribution $\mathcal{R}_\tau^\star$ when utterances are scored at the decision token. We have the following main theorem.

**Theorem C.1.** *Assume that the score $s$ is bounded between $[-M, M]$ for some constant $M > 0$. Let $\Omega(\{r_t\}_1^N) := \{\{r_t\} \in [-M, M]^N, r_1 \geq r_2 \geq \ldots \geq r_N\}$. Let $f(\cdot|r_k)$ be the PDF of $R_k|r_k$ and $F(\cdot|r_k)$ be its CDF. Let $f(\cdot)$ be the PDF of $r_k$, then the distribution of the optimal cutoff $\mathcal{R}_\tau^\star$ is computed as*

$$\mathbb{P}(\mathcal{R}_\tau^\star = k) = N! \int_{\Omega(\{r_t\}_1^N)} \left( \int_{-M}^{M} f(u|r_k) \prod_{[N] \backslash k} F(u|r_k) du \right) \prod_{t=1}^{N} f(r_t) \cdot dr_{1:N}. \ \forall k \in [N]$$

*Furthermore, $f(\cdot), f(\cdot|r), F(\cdot|r)$ can be fully expressed via the language model $p$ and the reward function $s$.*

*Remark* C.2. Our theorem indicates that when $r$ and $R$ has higher correlation, there is usually better rejection rate. For instance, consider the extreme case where the Pearson correlation between $r$ and $R$ is 1, then $R = C \cdot r + D$ for some

---

[1]Having access to the probability $p(\cdot|X)$ and reward function $s(X, \cdot)$ is also common in practice as we can query the logits of LLM for prompt $X$ and response $y$ and convert it to $p(y|X)$.

$C > 0$ and constant $D$. In this case, for $r_1 \geq r_2 \geq \ldots \geq r_N$,

$$\int_{-M}^{M} f(u|r_1) \prod_{[N]\backslash 1} F(u|r_k)du = 1, \quad \int_{-M}^{M} f(u|r_s) \prod_{[N]\backslash 1} F(u|r_k)du = 0, \; \forall s \in [N]\backslash 1.$$

Therefore, Theorem C.1 indicates $\mathbb{P}(\mathcal{R}_\tau^\star = 1) = 1$, i.e. we have the maximum cut-off that we can cutoff $N-1$ trajectories. Similarly, when $r$ and $R$ have strong positive correlation, i.e. $R = C \cdot r + D + \delta$ with random variable $\delta$ has small magnitude, then $\int_{-M}^{M} f(u|r_1) \prod_{[N]\backslash 1} F(u|r_k)du$ is also close to 1 since $\prod_{[N]\backslash 1} F(u|r_k)$ is close to 1 near the support of $R_1|r_1$.

*Proof of Theorem C.1.* **Step1.** We condition on the partial rewards $\{r_k\}_{k \in [N]}$ for the whole step. As a result, we can without loss of generality assume they are in the descending order, *i.e.*[2]

$$r_1 \geq r_2 \geq \ldots \geq r_N.$$

Given this, $R_k$ are generated from distributions $f(\cdot|r_k)$, where $f(\cdot|r_k)$ are PDFs of $R_k|r_k$. In particular, we denote the shorthand notation $f_k(\cdot) := f(\cdot|r_k)$. Let the maximal order statistics of $R_k$ be $R_{(1)} := \max_{k \in [N]} R_k$, then $\mathcal{R}_\tau^\star$ has the following equivalent mathematically expression:

$$\mathcal{R}_\tau^\star := \mathrm{argmin}_k \{k : R_k = R_{(1)}\},$$

Next, we compute the distribution of $\mathcal{R}_\tau^\star$ (recall we are conditioning on $r_k$'s so far). Indeed, note that, after module measure 0 event, we have $\mathcal{R}_\tau^\star = k \Leftrightarrow R_k = R_{(1)}$, therefore

$$\mathbb{P}(\mathcal{R}_\tau^\star = k|r_t, t \in [N]) = \mathbb{P}(R_k = R_{(1)}|r_t, t \in [N]) = \mathbb{P}(R_k = \max_{m \in [N]} R_m|r_t, t \in [N])$$

$$= \mathbb{P}(R_k \geq \max_{m \in [N]} R_m|r_t, t \in [N]) = \mathbb{P}(R_k \geq \max_{m \in [N]\backslash k} R_m|r_t, t \in [N])$$

$$= \int_{-M}^{M} \int_{-M}^{u} f_k(u) f_{\max_{m \in [N]\backslash k} R_m}(t)dtdu = \int_{-M}^{M} f_k(u) \left( \int_{-M}^{u} f_{\max_{m \in [N]\backslash k} R_m}(t)dt \right) du$$

$$= \int_{-M}^{M} f_k(u) F_{\max_{m \in [N]\backslash k} R_m}(u)du = \int_{-M}^{M} f_k(u) \prod_{[N]\backslash k} F_k(u)du.$$

Here the fifth equal sign uses $R_k$ is independent of $\max_{m \in [N]\backslash k} R_m$ given $r_{1:N}$, and $f_{\max_{m \in [N]\backslash k} R_m}$ is the PDF of $\max_{m \in [N]\backslash k} R_m|r_{1:N}$. The last equal sign uses Lemma C.4.

**Step2:** Now we relax the condition on $r_k$'s and allow them to arbitrarily ordered. Recall $r_k$'s are i.i.d. with PDF $f(\cdot)$. In this case, we can always list its corresponding order statistics as:

$$r_{(1)} \geq r_{(2)} \geq \ldots r_{(N)},$$

and the joint PDF $f_{\mathrm{joint}}$ for $(r_{(1)}, r_{(2)}, \ldots, r_{(N)})$ is

$$f_{\mathrm{joint}}(r_1, \ldots, r_N) := f_{(r_{(1)}, r_{(2)}, \ldots, r_{(N)})}(r_1, \ldots, r_N) = N! \cdot \prod_{k=1}^{N} f(r_k) \cdot \mathbf{1}_{\{r_1 \geq r_2 \geq \ldots \geq r_N\}}.$$

In this setup, what we obtained in Step1 is actually $\mathbb{P}(\mathcal{R}_\tau^\star = k|r_{(t)}, t \in [N])$. Therefore, via law of total probability, we have

$$\mathbb{P}(\mathcal{R}_\tau^\star = k) = \int_{\{r_t\} \in [-R,R]^N} \mathbb{P}(\mathcal{R}_\tau^\star = k|r_{(t)} = r_t, t \in [N]) \cdot f_{\mathrm{joint}}(r_{1:N}) \cdot dr_{1:N}$$

$$= \int_{\{r_t\} \in [-R,R]^N, r_1 \geq r_2 \geq \ldots r_N} N! \cdot \mathbb{P}(\mathcal{R}_\tau^\star = k|r_t, t \in [N]) \cdot \prod_{t=1}^{N} f(r_t)dr_{1:N}$$

$$= N! \int_{\{r_t\} \in [-R,R]^N, r_1 \geq r_2 \geq \ldots r_N} \left( \int_{-M}^{M} f(u|r_k) \prod_{[N]\backslash k} F(u|r_k)du \right) \cdot \prod_{t=1}^{N} f(r_t)dr_{1:N}.$$

---

[2]In the next step, we will relax this condition.

Here $f(\cdot|r_k)$ and $f(\cdot)$ are explicitly computed via Lemma C.3 and $F(u|r_k) = \int_{-R}^{u} f(z|r_k)dz$. The last equal sign uses Step1. This finish the proof.

$\square$

**Lemma C.3.** *Recall $r_k$ is fixed and $f_k(r) = f(r|r_k) = \mathbb{P}(R_k = r|R_k^\tau = r_k)$. Then $f_k$ can be explicitly expressed by $p, s, \tau, X$ via*

$$f_k(r) = \frac{\sum\limits_{y \in s^{-1}(r|X, y^\tau)} \sum\limits_{y^\tau \in s^{-1}(r_k|X)} p(y|X, y^\tau) \cdot p(y^\tau|X)}{\sum_{y^\tau \in s^{-1}(r_k|X)} p(y^\tau|X)}$$

*and*

$$f(r) = \sum_{y^\tau \in s^{-1}(r|X)} p(y^\tau|X).$$

*Here $s^{-1}(r|X) := \{y : s.t. \ s(X, y) = r\}$ and $s^{-1}(r|X, y') := \{y : s.t. \ s(X, y) = r, \ y^\tau = y'\}$ are (conditioning) level sets calculated via reward function $s$.*

*Proof.* Indeed,

$$f_k(r) = \mathbb{P}(R_k = r|R_k^\tau = r_k) = \mathbb{P}(s(X, y) = r|s(X, y^\tau) = r_k)$$

$$= \mathbb{P}[y \in s^{-1}(r|X, y^\tau)|y^\tau \in s^{-1}(r_k|X)] = \frac{\mathbb{P}[y \in s^{-1}(r|X, y^\tau), y^\tau \in s^{-1}(r_k|X)]}{\mathbb{P}[y^\tau \in s^{-1}(r_k|X)]}$$

$$= \frac{\sum\limits_{y \in s^{-1}(r|X, y^\tau), y^\tau \in s^{-1}(r_k|X)} p(y|X, y^\tau)p(y^\tau|X)}{\sum_{y^\tau \in s^{-1}(r_k|X)} p(y^\tau|X)}$$

which is fully characterized by $p, s, \tau, X$. Here $s^{-1}(r|X) := \{y : s.t. \ s(X, y) = r\}$ and $s^{-1}(r|X, y') := \{y : s.t. \ s(X, y) = r, \ y^\tau = y'\}$ are (conditioning) level sets described by reward function $s$. Similarly, we have

$$f(r) = \mathbb{P}[s(X, y^\tau) = r] = \mathbb{P}[y^\tau \in s^{-1}(r|X)] = \sum_{y^\tau \in s^{-1}(r|X)} p(y^\tau|X).$$

$\square$

**Lemma C.4.** *Let $X_k$'s ($1 \leq k \leq N$) be the independent random variables with respective cumulative distribution function (CDF) $F_k$. Then the CDF of $\max_{k \in [N]} X_k$ is $\prod_{k=1}^{N} F_{X_k}(x)$.*

*Proof.* Indeed, by independence

$$F_{\max_k X_k}(x) = \mathbb{P}(\max_k X_k \leq x) = \prod_{k=1}^{N} \mathbb{P}(X_k \leq x) = \prod_{k=1}^{N} F_{X_k}(x).$$

$\square$

## D. Correlation between partial and final rewards

In this section, we present our observation that the partial and final rewards are positively correlative for the responses to a single prompt. We examine the distribution for the (empirical) Pearson correlation and Kendall's tau correlation coefficient for partial and final rewards for a single prompt. Mathematically, for $(X_1, X_2, ..., X_N)$ and $(Y_1, Y_2, ..., Y_N)$, the two correlation are defined as

$$R_{\mathsf{Pearson}} := \frac{\sum_{i=1}^{N}(X_i - \bar{X})(Y_i - \bar{Y})}{\sqrt{\sum_{i=1}^{N}(X_i - \bar{X})^2 \cdot \sum_{i=1}^{N}(Y_i - \bar{Y})^2}},$$

$$R_{\mathsf{Kendall}} := \frac{2}{N(N-1)} \sum_{i<j} \mathsf{sgn}(X_i - X_j) \cdot \mathsf{sgn}(Y_i - Y_j),$$

where $\bar{X} = \sum_{i=1}^{N} X_i/N, \bar{Y} = \sum_{i=1}^{N} Y_i/N$ are their average, and $\text{sgn}(\cdot)$ is the sign function.

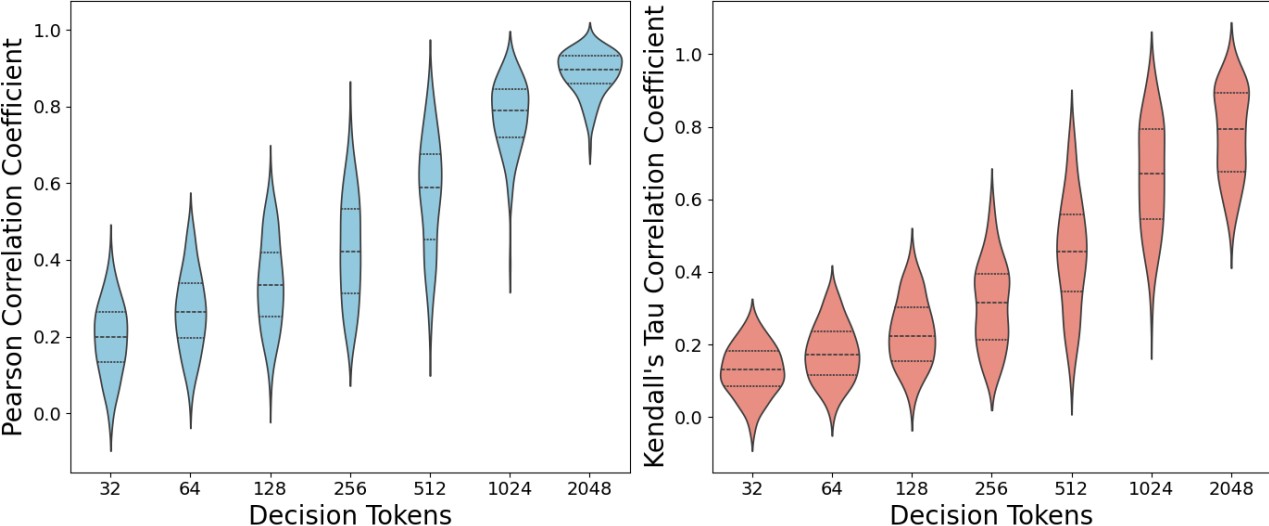

*Figure 5.* Pearson correlation (left) and Kendall's tau correlation coefficient (right) for the partial and final rewards. We randomly sample 100 prompts in the AlpacaFarm-Eval dataset. The responses are generated via Llama3-8b–Instruct and rewards are evaluated via Mistral-7B-RM.

## E. More results in the counterfactual analysis

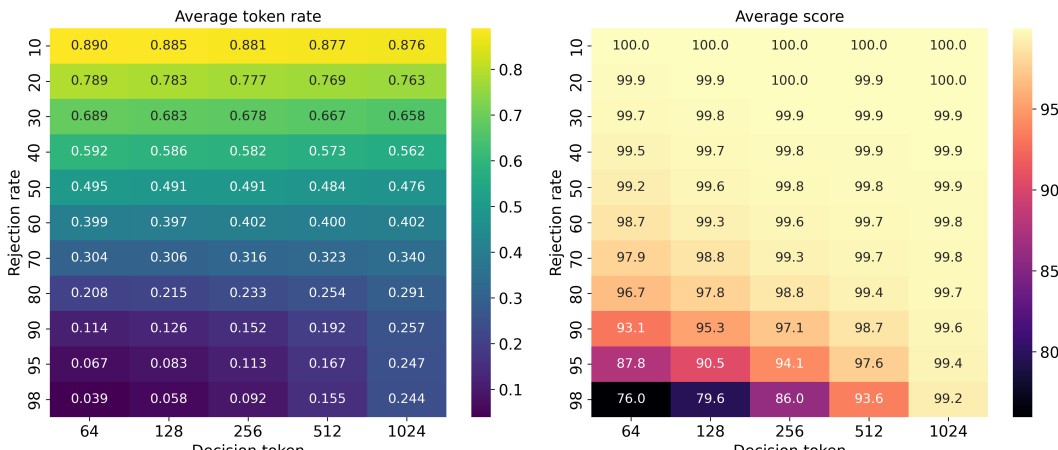

*Figure 6.* Counterfactual analysis for the 100 prompts in AlpacaFarm-Eval set. Responses are generated via Llama3-8B-Instruct, and rewards are evaluated via Mistral-7B-RM.

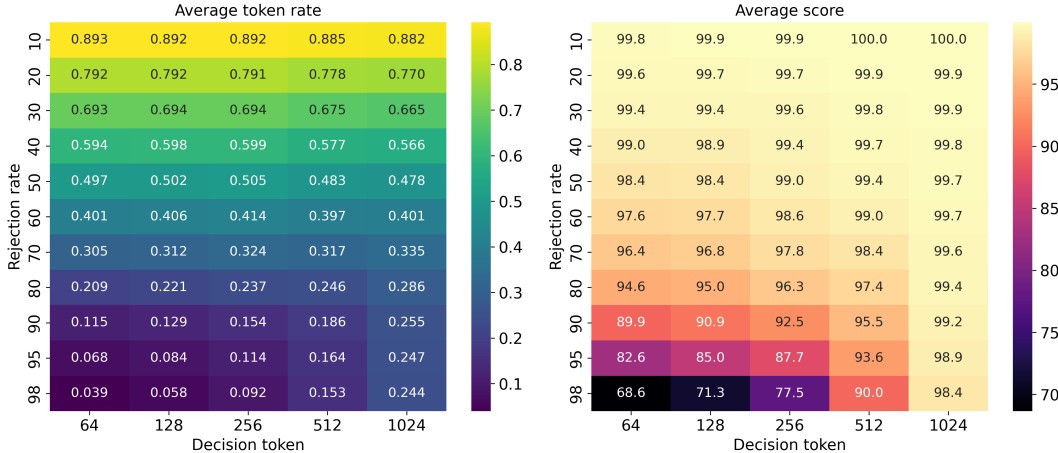

*Figure 7.* Counterfactual analysis for the 100 prompts in AlpacaFarm-Eval set. Responses are generated via Llama3-8B-Instruct, and rewards are evaluated via Eurus-7B-RM.

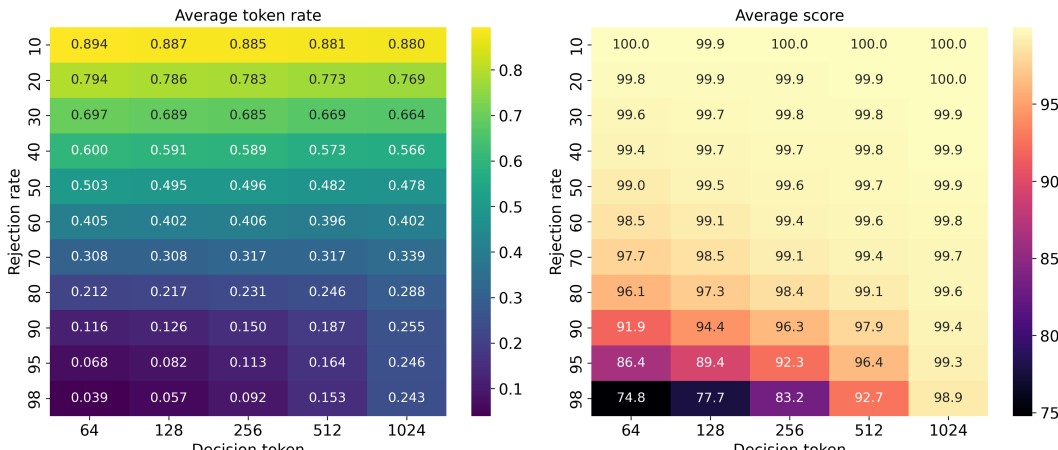

*Figure 8.* Counterfactual analysis for the 100 prompts in AlpacaFarm-Eval set. Responses are generated via Llama3-8B-Instruct, and rewards are evaluated via Llama3-8B-RM.

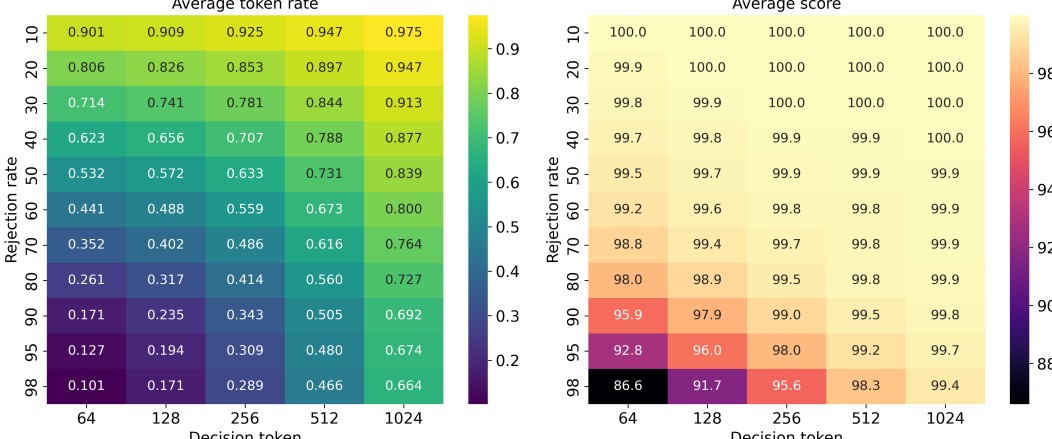

*Figure 9.* Counterfactual analysis for the 100 prompts in AlpacaFarm-Eval set. Responses are generated via Llama3-8B, and rewards are evaluated via Mistral-7B-RM.

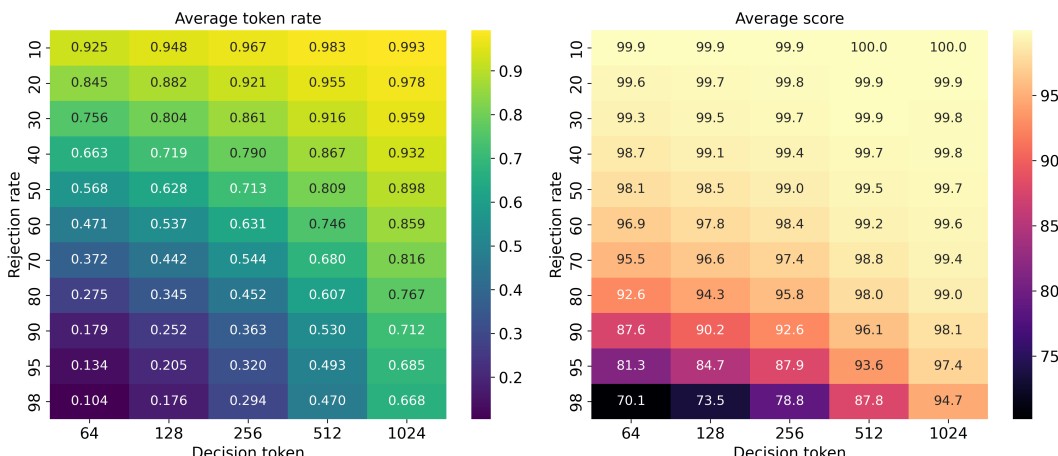

*Figure 10.* Counterfactual analysis for the 100 prompts in AlpacaFarm-Eval set. Responses are generated via Llama3-8B, and rewards are evaluated via Eurus-7B-RM.

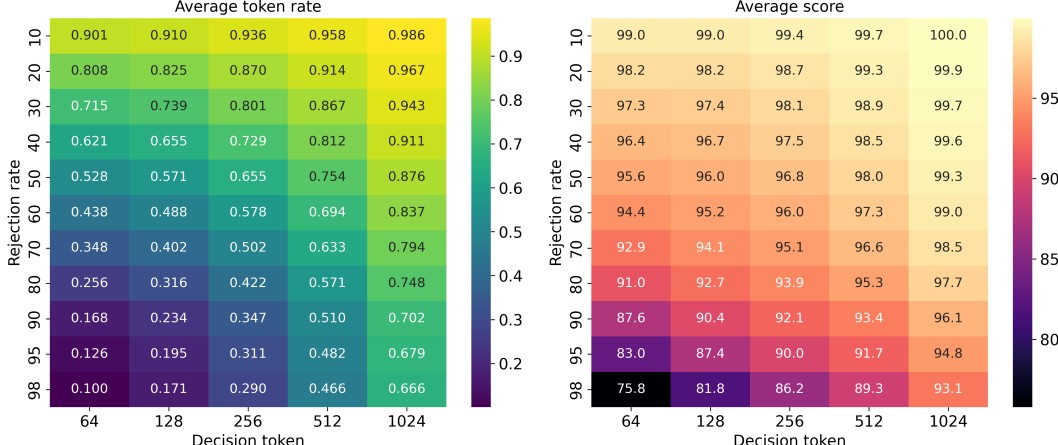

*Figure 11.* Counterfactual analysis for the 100 prompts in AlpacaFarm-Eval set. Responses are generated via Llama3-8B, and rewards are evaluated via Llama3-8B-RM.

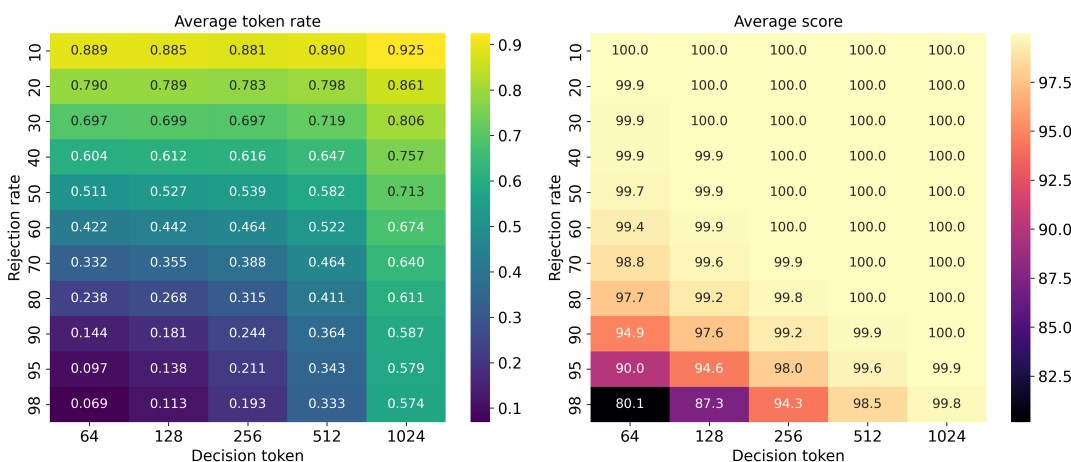

*Figure 12.* Counterfactual analysis for the 100 prompts in AlpacaFarm-Eval set. Responses are generated via AF-SFT10K, and rewards are evaluated via Mistral-7B-RM.

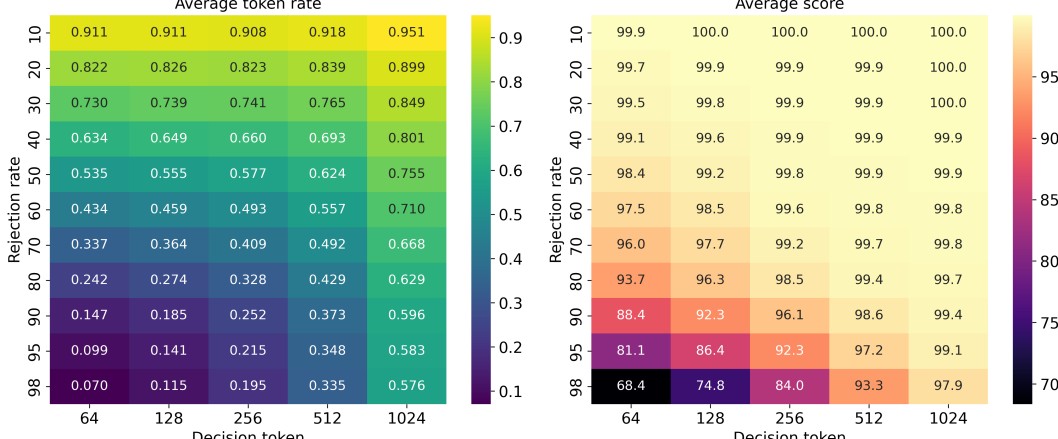

*Figure 13.* Counterfactual analysis for the 100 prompts in AlpacaFarm-Eval set. Responses are generated via AF-SFT10K, and rewards are evaluated via Eurus-7B-RM.

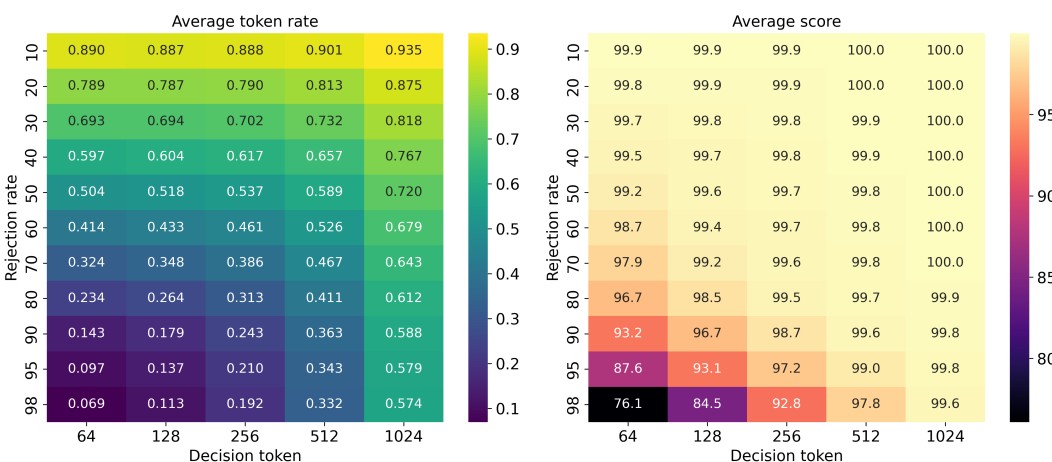

*Figure 14.* Counterfactual analysis for the 100 prompts in AlpacaFarm-Eval set. Responses are generated via AF-SFT10K, and rewards are evaluated via Llama3-8B-RM.

