# OpenReview forum: "Accelerating Best-of-N via Speculative Rejection"
_ICML.cc/2024/Workshop/WANT — WANT@ICML 2024 Poster_

### Official Review · Reviewer_F3gB · 2024-06-12
**A simple yet efficient method to speed-up the Best-of-N algorithm without significantly degrading the scores of the selected generated texts.**

**Confidence:** 3

**Summary:**

This work proposes a simple heuristic to efficiently sample high scoring text generations according to the best-of-N algorithm. The method---called Speculative Best-of-N (SBoN)---consists in (i) generating N completions of $\tau$ tokens in parallel, then (ii) computing a rejection threshold $r_{cut}$ as a the $\alpha$-th lower quantile of rewards, and (iii) continue generating up to completion all the sequences with a score larger than $r_{cut}$, score those, and return the argmax.

They demonstrate that their method can be motivated by the correlation between partial and full scores. They show how SBoN is competitive when compared to the baseline BoN method, while being faster.

**Strengths:**

The method is well motivated, efficient text generation is an important research topic.

The paper is easy to follow.

The method is simple to implement.

I find the results compelling: the speedup is significant and the SBoN scores are competitive compared to Bo100.

I like the comparison to BoM, even though I am confused by how M is selected (see weaknesses).

**Weaknesses:**

The novelty of the method is low, but this doesn't mean it cannot be valuable to the community.

The correlation measured for smaller $\tau$ values is relatively small. It might be that the algorithm performs well not entirely due to the correlation but to the large sampling size which guarantees to find at least one sequence with a large partial score and a large final score. I see in the appendix you also show results for N=50, yet the SBoN scores are larger than 100, I am unsure how this can happen?

In my understanding, the BoM baseline should have the same speedup as SBoN, I do not understand why this is not the case.

Given the relatively weak correlation, SBoN introduces a bias in the generation. It could be interesting to provide some of the high scoring generated text which got selected by BoN and rejected by SBoN.

**Limitations:**

See weaknesses

**Suggestions:**

See weaknesses

---

### Official Review · Reviewer_MHry · 2024-06-13
**Unsupported assumptions; weak experiments; vague presentation; inappropriate typesetting.**

**Confidence:** 5

**Summary:**

The authors proposed a method for reducing generation time with LLMs in Best-of-N generation strategy. The authors use word "speculative" to explain main idea of trading of batch generation and rejecting (or pruning) of completions on early staged with reward model.

**Strengths:**

+ The work introduces interesting perspective with impact on practical application.

+ In the appendix C, some theoretical guarantee described.

**Weaknesses:**

## Typesetting

There are shortcommings in the draft typesetting. From my perspective, the
paper should be desk rejected since there are serious ones.

- Abstract consists of two paragraphs of differnt style. Specifically, the
  second paragraph has font size of main body which is larger than required
  font size in abstract. The same true for leading and paragraph spacing.
- Bibliography should be reviewed and actualized:  capitalization of titles,
  missing publication dates, journals conferences, etc (e.g. RAFT is published in
  #link("https://openreview.net/forum?id=m7p5O7zblY")[TMLR]).
- Missing table of content in hypertext markup.
- Intoductory section seems incoherent and hard to follows.
- It would be better render figure 2, 3, and 4 as vector graphics. There is a
  light noticable lag during scrolling page 8.
- Please introduce common abbreviation as early as possible. For example, "A
  popular and well known decoding strategy for this purpose is the Best-of-N
  (BoN) method".

## Major Points of Criticism

### Speculative Rejection

+ Term "speculative rejection" is not defined in section "Speculative Rejection".

+ Statement about correlation between final reward and partial reward seems
  unconvincing. Specifically, Figure 2 looks a little bit strange. I would say
  that the blue line should be located more in the point cloud (e.g. $y = 4 x -
  4x$).

  Also, this experiment reveals correlation between one specific prompt. It
  makes me doubt about generality of your claim. It would better to take
  statistically significant number of prompts and generate multiple
  continuations for each prompt. Optionally, this experiment should be repeated
  for other pair of language model and reward model.

  I'd like suggest authors to dig in this idea deeper and study this
  monotonicity property more thorough. It seems fruitfull finding which might
  result in other applications.

### Experiment

+ Comparison among different BoN methods are solid. However, there is a lack of
  comparison with other speculative generation techniques. They give
  approximately the same speed up thus it is unclear wheter the proposed method
  is competitive.

+ Also, trivial generalization of the proposed method is not studied. For
  example, there is only one threshold time $tau$. Why not two? May be series
  of exponentially spaced $tau_k$ is the best?

### Efficiency

There is no discussion of joint use of speculative rejection and speculative
decoding techniques (e.g. (Leviathan, 2022)). It is obviously possible to use
them simulteneously but some engineering issues related to block size and batch
raise. Also, this issue opens another question related to sharing weights in
draft (surrogate) model and reward model.

## Others

### Related Works

+ There are unsuitable references. For example in *Inference Efficiency in
  LLMs*. QLoRA is a PEFT + quantization for training. Correct general examples
  of quantization (PQT) is AQLM and QUIP\# which are SOTA post-training
  quantization techniques. There are other quantization approaches which
  requires training (AQT) or quantization of KV-caches for efficient inference.
+ Reference to vLLM is not enought since there are other competitive techniques
  like DeepSpeed and TensorRT.

### Speculative Best-of-N (SBoN)

+ There is missing definition for $cal(l)_k$ (missing in main body and unclear
  use in Algo.1)

### Problem Formulation

+ Please rewrite Best-of-N optimization problem as a block equation. Now it
  looks quite blurry while it is a short meaningful formal definition. Also,
  use the same notation throughout the text (e.g. symbols $=$ and $:=$ are used

### Appendix

+ I think that the theoretical guarantee described in the appendix must be
  presented in the main body at least as the theorem C.1

+ From my perspective, there are too many figures in the appendix. It is hard
  to follow each of them.

---

### Official Review · Reviewer_Fe7c · 2024-06-16
**Interesting work with some limitations**

**Confidence:** 3

**Summary:**

This paper presents a novel technique to speed up the Best-of-N method for decoding with early stopping of some of the utterances which they pre-maturely identity as wrong or undesirable. The technique seems pretty straightforward and useful. The technique is called speculative rejection and share some overall with speculative decoding and beam search decoding.

**Strengths:**

Some major strengths are listed below.

* This is a well written paper.
* The problem, motivation and proposed method is very clearly stated and makes intuitive sense.
* This results are shown with reasonably large sized models.

**Weaknesses:**

Although I would like to emphasis that this is a good paper but it has some limitations which if addressed this could be great paper.

* Although the paper says that it is trying to speed up BoN and compares only with BoN and its derivatives as baselines I believe some more non BoN type baseline can helpful to see overall efficacy of this method. Overall I believe more relevant baselines are required.

* We know that KV caching helps in generation speedups, since the core claim of this paper is to faster generations, I would like to request authors to make some non decoding method as baseline as well and compare against it or simply show that it can further improve the speed of generation.

* The authors have not discussed about the quality of prompts i.e. does this method works for hard prompts? A new table can be added to show efficacy of this method for easy, medium and hard prompts.

**Limitations:**

Already written as weakness.

**Suggestions:**

If the weaknesses are addressed I think this could be a great paper.

---

### Decision · Program_Chairs · 2024-06-18

**Decision:**

Accept (Poster)

**Comment:**

We thank the authors for their time and contribution to WANT and we are pleased to share that after the reviewing process the paper has been accepted. Congratulations! We encourage the authors to consider reviewers' feedback for the improvement of the camera-ready version. We hope to see you in person at the workshop and brainstorm on efficient training research together!